# MaterialRefGS: Reflective Gaussian Splatting with Multi-view Consistent Material Inference

**Wenyuan Zhang**[1][*]    **Jimin Tang**[1][*]    **Weiqi Zhang**[1]    **Yi Fang**[2]
**Yu-Shen Liu**[1][†]    **Zhizhong Han**[3]

School of Software, Tsinghua University, Beijing, China[1]
Center for AI and Robotics (CAIR), NYU Abu Dhabi, UAE[2]
Department of Computer Science, Wayne State University, Detroit, USA[3]
{zhangwen21,tangjm24,zwq23}@mails.tsinghua.edu.cn
yfang@nyu.edu    liuyushen@tsinghua.edu.cn    h312h@wayne.edu

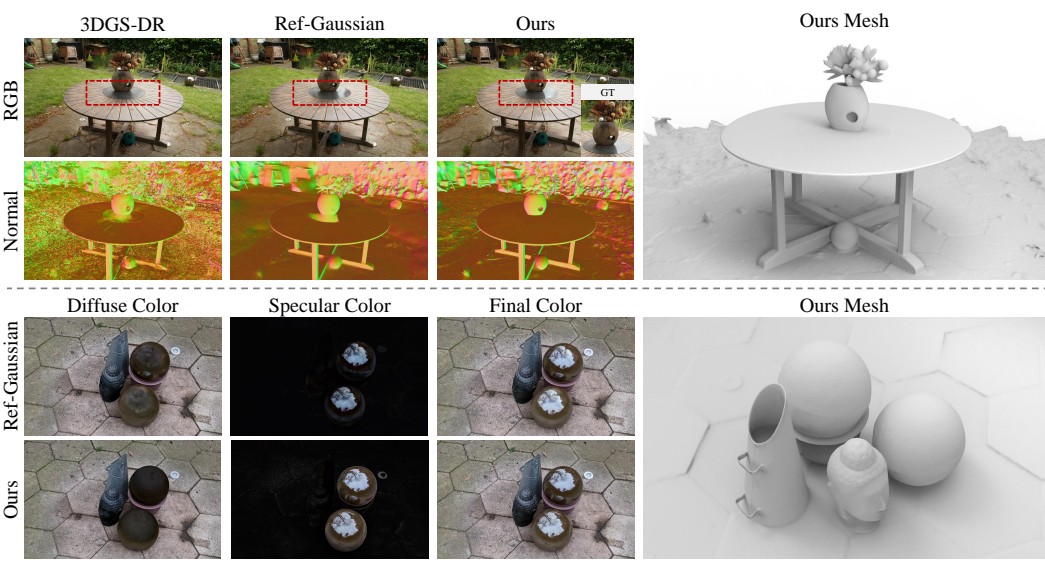

Figure 1: We highlight our novel view synthesis results on real-world scenes with complex reflection effects. Our **MaterialRefGS** outperforms the state-of-the-art methods in producing photorealistic renderings, disentangling physical materials, and recovering accurate scene geometry.

## Abstract

Modeling reflections from 2D images is essential for photorealistic rendering and novel view synthesis. Recent approaches enhance Gaussian primitives with reflection-related material attributes to enable physically based rendering (PBR) with Gaussian Splatting. However, the material inference often lacks sufficient constraints, especially under limited environment modeling, resulting in illumination aliasing and reduced generalization. In this work, we revisit the problem from a multi-view perspective and show that multi-view consistent material inference with more physically-based environment modeling is key to learning accurate reflections with Gaussian Splatting. To this end, we enforce 2D Gaussians to produce multi-view consistent material maps during deferred shading. We also track photometric variations across views to identify highly reflective

---

[*]Equal contribution.
[†]The corresponding author is Yu-Shen Liu.

39th Conference on Neural Information Processing Systems (NeurIPS 2025).

regions, which serve as strong priors for reflection strength terms. To handle indirect illumination caused by inter-object occlusions, we further introduce an environment modeling strategy through ray tracing with 2DGS, enabling photorealistic rendering of indirect radiance. Experiments on widely used benchmarks show that our method faithfully recovers both illumination and geometry, achieving state-of-the-art rendering quality in novel views synthesis. Project Page: https://wen-yuan-zhang.github.io/MaterialRefGS.

## 1 Introduction

Learning scene appearance representations and recovering unseen views from multiple posed RGB images has been a long-standing task in computer vision and graphics [22, 36, 18, 64]. Recent advances in Neural Radiance Fields (NeRF) [36] leverage volume rendering to learn implicit scene representations for novel view synthesis. More recent efforts have been made by learning explicit 3D Gaussians (3DGS) [22] to achieve real-time rendering through a differentiable splatting procedure. Despite achieving photorealistic synthesis, 3DGS shows limited performance when confronted with complex reflective environments. This limitation arises from the contradiction between the simplistic geometric representations and the intricate shading mechanisms of real-world objects.

To tackle this problem, recent methods typically separate the rendering color into diffuse and specular components, and adopt inverse rendering frameworks [55, 11] to learn illumination decomposition through physically based rendering [40]. They endow each Gaussian primitive with learnable reflection-related properties, such as metallic and roughness [20, 24]. A splatting pass rasterizes these attributes into screen-space material maps [22, 18], followed by a lighting pass that evaluates a Bidirectional Reflectance Distribution Function (BRDF) [4] using material maps and environment lighting to synthesize the final image. This two-stage pipeline is known as deferred shading-based PBR [24, 56, 66]. It decomposes the view-dependent reflection effects into view-independent material properties by considering how the light interacts with the objects and environments, thereby improving the fidelity of novel views. However, the illumination decomposition poses several optimization challenges. First, inferring material properties from multi-view images is an ill-posed problem. All material parameters are optimized only through photometric loss after complex light transport, so multiple combinations of lighting and materials can explain the same pixels, which often leads to suboptimal illumination decomposition [46, 58]. Second, the view-dependent behavior of 3D Gaussian representations conflicts with the goal of learning view-independent material properties. When the same physical attribute yields inconsistent appearances in different viewpoints, the BRDF struggles to infer accurate reflections from ambiguous observations, resulting in aliasing and degenerated illumination decomposition [10, 31].

To resolve these issues, we propose a novel approach that learns illumination decomposition for modeling reflections with 2DGS through multi-view consistent material inference. We first enforce Gaussians to produce multi-view consistent material buffers based on their physical attributes. This is achieved by aligning the projections of geometric surfaces on material maps from different views. We find that this constraint significantly improves illumination decomposition by limiting the view-specific overfitting. To better facilitate this process, we track photometric variations on object surfaces along the camera trajectory and quantify these variations as reflection scores. A spatial reflection fusion module is then applied to aggregate these per-view reflection scores into a multi-view consistent reflection strength prior. This prior is subsequently used as a supervision for the reflection strength attribute, i.e., metallic.

In addition, we observe that secondary reflection effects caused by inter-object occlusions often lead to degraded novel view synthesis. To address this, we propose an improved environment modeling strategy via differentiable ray tracing, which combines splatted indirect radiance and queried direct radiance with an on-the-fly estimation of occlusion probability. This approach effectively provides physically grounded signals in occluded regions, enabling more realistic indirect illumination. Our numerical and visual evaluations on widely used benchmarks demonstrate our superiority over the latest methods in terms of material inference and novel view synthesis. Our contributions are summarized as follows:

- We propose a novel approach to modeling reflections through Gaussian Splatting with multi-view consistent material inference, including multi-view material consistency constraint and

reflection strength prior supervision. Our approach provides a new perspective for modeling reflections through physically grounded illumination decomposition.

- We introduce a differentiable environment modeling strategy through 2DGS based ray tracing, which enhances photorealistic rendering of indirectly illuminated regions caused by inter-object occlusions.

- We achieve state-of-the-art performance of novel view synthesis both in numerical results and visual comparisons on widely used benchmarks.

## 2 Related Work

### 2.1 Novel View Synthesis

The task of novel view synthesis aims to predict unseen views of a scene from a set of posed RGB images. Traditional methods typically rely on image interpolation [45] or inpainting [2] to generate novel views. With the rapid development of deep learning [70, 69, 67, 68, 5, 39, 28, 52, 32], novel view synthesis has gradually shifted toward learning-based approaches. Neural Radiance Fields (NeRF) [36, 62, 63, 65, 16, 51] pioneers this task by learning a mapping from 5D coordinates to volume densities as the scene representations. More recently, 3D Gaussian Splatting (3DGS) [22] has emerged as a new paradigm for real-time rendering by rasterizing Gaussian ellipsoids into images in a splatting manner. Various extensions support diverse scales and scenes through novel data structures such as hierarchies [23, 34] and octrees [44]. Others address sparse-view challenges by incorporating geometric priors [7, 19, 27, 17]. Beyond static scenes, some works also explore 3DGS for dynamic scenes [50], semantic-aware manipulation [42], and content generation [71, 60, 9]. Recent efforts aim to extract high-quality surfaces from 3DGS by flattening 3D Gaussians into 2D disks [18] and leveraging differentiable kernels to rasterize them into images. To better align Gaussians with object surfaces, regularization strategies such as depth-normal consistency [18, 15] and neural gradient supervision [61, 29, 35] are applied. In our work, we adopt 2D Gaussians as the foundational representation due to their effectiveness in modeling surface geometry and normals.

### 2.2 Modeling Reflections in NeRF and 3DGS

The view-dependent color representations used in original NeRF [36] and 3DGS [22], such as neural networks or spherical harmonics, struggle to capture high-frequency specular reflections that are commonly observed in real-world scenes. Existing solutions typically decompose the outgoing radiance into diffuse and specular components and blend them using learnable weights. To better model the specular reflections, some methods introduce directional encodings like Integrated Directional Encoding [47] and Gaussian Directional Encoding [26]. Other approaches extract accurate meshes to provide reliable normal for reflection modeling [33, 48, 13, 25]. Recent advances in 3DGS [22] offer new perspectives for addressing this challenge. Inspired by inverse rendering, Relight3DGS [11] assigns each Gaussian with physical properties such as metallic and roughness, and performs PBR on the Gaussians to synthesize the final image. Recent studies have proven that rendering per-Gaussian illumination attributes into material maps followed by deferred shading-based PBR [24, 56, 66, 20] yields better performance than per-Gaussian shading [11]. To improve environment interactions, some methods develop ray tracing techniques for Gaussians [37, 14, 53]. However, these methods primarily focus on per-view light-material interactions and neglect the globally consistent geometric information inherent in the multi-view settings. To fill in this gap, we propose leveraging multi-view cues to facilitate the disentanglement of material properties, enabling more accurate and realistic modeling of reflections in 3D Gaussians. Notably, our method is not equivalent to inverse rendering. Our goal is to model specular color through illumination decomposition, while relying on the Gaussians to render the diffuse color. In contrast, inverse rendering methods evaluate all lighting effects through BRDF, making them more suitable for quantitatively evaluating material decomposition and for relighting tasks. However, due to the complexity of learning diffuse component, these methods are limited to simple object-centric scenes.

## 3 Method

Given a set of posed RGB images $\{I_j\}_{j=1}^{N}$ that represents a scene with high reflections, we aim to synthesize a novel image from an unseen viewpoint. We learn a set of 2D Gaussians as the scene

representations. We begin by introducing the preliminaries (Sec. 3.1), and then describe our multi-view material inference strategy (Sec. 3.2) and environment modeling strategy (Sec. 3.3). Finally, we detail the optimization procedure (Sec. 3.4). An overview of our method is provided in Fig. 2.

## 3.1 Preliminary

3D Gaussian Splatting (3DGS) [22] has become paradigms for learning 3D representations from multi-view images. A scene is represented by Gaussian functions $\{G_i\}_{i=1}^{K}$ with attributes like mean $x_i$, opacity $o_i$ and scaling $s_i$. We also attach several reflection-related material attributes to Gaussians, including diffuse color $c_d \in \mathbb{R}^3$, albedo $a \in \mathbb{R}$, metallic $m \in \mathbb{R}$ and roughness $r \in \mathbb{R}$. We can then rasterize these Gaussians into images using

$$\hat{\Psi} = \sum_{i=1}^{N} \psi_i * o_i * p_i * \prod_{j=1}^{i-1}(1 - o_j), \tag{1}$$

where $o_i, p_i$ are the opacity and screen-space probability [72] of the $i$-th Gaussian, respectively, and $\psi_i$ denotes a selected attribute of $G_i$. By choosing different attributes such as $c_{di}, a_i, m_i$ or $r_i$ as $\psi_i$, we can render the corresponding material maps $\Psi^{C_d}, \Psi^A, \Psi^M, \Psi^R$, respectively. To facilitate surface reconstruction, 2DGS [18] flattens each 3D Gaussian into a 2D disk by setting one scaling dimension to zero. We adopt 2DGS as our base representation for better surface and normal alignment.

Our deferred shading-based PBR adopts a simplified version of the Disney BRDF model [4]. Given a viewing direction $\omega_o$, the rendered color on the ray-surface intersection can be computed by

$$c(\omega_o) = (1 - m)c_d + L_s(\omega_o, a, m, r, n),$$
$$L_s(\omega_o, a, m, r, n) = \int_{\Omega} L_i(\omega_i)f_s(\omega_i, \omega_o)(\omega_i \cdot n)d\omega_i, \tag{2}$$
$$f_s(\omega_i, \omega_o) = \frac{DFG}{4(\omega_i \cdot n)(\omega_o \cdot n)}$$

where $\omega_i, n, L_s, f_s, D, F, G$ denote the incident direction, normal, outgoing specular radiance, BRDF term, normal distribution function, Fresnel term and shadowing-masking term, respectively. Since computing the integral of $L_s$ over the upper hemisphere $\Omega$ is computationally expensive, we adopt the split-sum approximation [38, 56, 31], which separates the integral into two components,

$$L_s(\omega_o, a, m, r, n) \approx \int_{\Omega} f_s(\omega_i, \omega_o)(\omega_i \cdot n)d\omega_i \cdot \int_{\Omega} L_i(\omega_i)D(\omega_i, \omega_o)(\omega_i \cdot n)d\omega_i, \tag{3}$$

where the first term can be precomputed using $a, m, r$ and stored in a look-up table. The second term can be queried from a set of learnable environment cubemaps using reflected direction and $r$.

Similar to reflective Gaussian methods [20, 56], we decouple the diffuse component $c_d$ and the specular component $L_s$, assigning the prediction of $c_d$ to Gaussian rasterization. Unlike classical graphics pipelines that jointly infer diffuse and specular terms from albedo and roughness [31, 14], we find that such a design significantly increases optimization difficulty, especially in complex real-world scenes. Moreover, since the diffuse component is relatively insensitive to viewing direction, delegating its prediction to the Gaussians allows us to better disentangle reflection effects from illumination.

## 3.2 Multi-view Consistent Material Inference

Current methods model scene reflections by learning a set of Gaussians associated with material properties through PBR. The underlying assumption is that view-dependent specular variations can be disentangled into view-independent material attributes, while evaluating the final view-dependent appearance can be deputed to BRDF. However, this assumption often breaks down in practice, as shown in Fig. 8 (a), where the learned material maps exhibit significant discontinuities and inconsistency across different views. Since Gaussians cover different pixels and contribute varying weights across viewpoints during alpha blending, material parameters exhibit significant inconsistency from different perspectives. This inconsistency hampers accurate illumination decomposition, as the BRDF struggles to infer a global physical reflectance effect from such inconsistent material

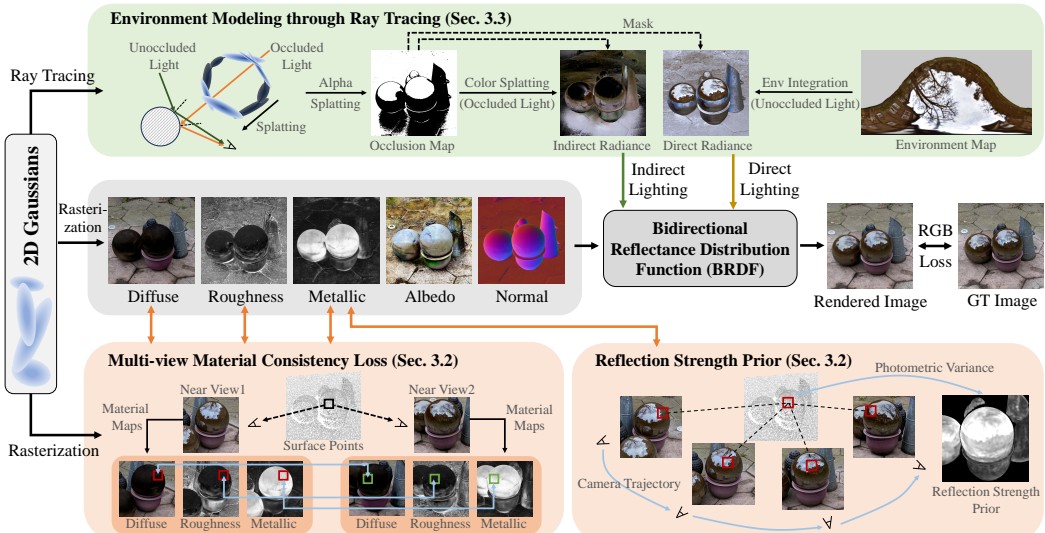

Figure 2: Overview of our method. We learn illumination decomposition by imposing multi-view material consistency constraint and reflection strength prior supervision on the rasterized material maps (Sec. 3.2). To facilitate this process, we introduce an environment modeling strategy through ray tracing with 2DGS, which effectively captures photorealistic incident lighting effects (Sec. 3.3).

observations. To overcome this obstacle, we propose to learn illumination decomposition by exploring multi-view consistent geometric clues as material inference constraints and guidance.

**Multi-view Material Consistency.** Based on the above analysis, we constrain the 2D Gaussians to produce multi-view consistent material maps, which is essential for accurate light-object-environment interactions inference. Specifically, for a surface point $p$ visible from both viewpoint $v_i$ and $v_j$, we want the projection $\pi(p)$ of $p$ on the two material maps $\Psi_i(\pi_i(p)), \Psi_j(\pi_j(p))$ to be the same. Drawing inspiration from multi-view stereo methods [8, 6], we impose constraints on plane patches between adjacent views. We first sample a $7 \times 7$ pixel patch $P(\pi_i(p))$ around $\pi_i(p)$, back-project it into 3D space along $v_i$ using the rendered depth $d_i$ at $\pi_i(p)$, rotate it with the rendered normal at $\pi_i(p)$, and reproject it into $\Psi_j$ using the rendered depth at $\pi_j(p)$, to form a warped patch $P'(\pi_j(p))$,

$$P'(\pi_j(p)) = H_{ij}P(\pi_i(p)), \quad H_{ij} = K_j(R_{ij} - \frac{T_{ij}n_i^T}{d_i})K_i^{-1}, \tag{4}$$

where $K_i, K_j$ are the intrinsic parameters of camera $v_i$ and $v_j$, $R_{ij}, T_{ij}, H_{ij}$ denote the relative rotation, relative translation and homography matrix from $v_i$ to $v_j$, respectively. We then enforce consistency between the two patches on the material maps using an MSE loss,

$$\mathcal{L}_{mv} = \|\Psi_i[P(\pi_i(p))] - \Psi_j[P'(\pi_j(p))]\|_2. \tag{5}$$

In practice, we once select one reference view along with multiple source views, warp the patch from the reference view to each source view, and compute a loss for every reference–source patch pair. The multi-view material consistency constraint is imposed to diffuse, roughness and metallic maps $\Psi^{C_d}, \Psi^R, \Psi^M$. We do not use this constraint on the albedo map $\Psi^A$, since albedo contributes little to most non-reflective surfaces, and enforcing consistency is ineffective and potentially harmful.

**Multi-view Consistent Reflection Strength Prior.** Multi-view consistency on material maps alone is insufficient to provide clear guidance for illumination decomposition. Based on the observation that highly reflective surfaces exhibit significantly different appearances across different viewpoints [3, 13], we explore multi-view photometric variations as explicit supervision for reflection strength. We first apply luminance normalization [54] on the ground truth RGB images to eliminate brightness inconsistencies caused by shadows and textures. As illustrated in Fig. 3, given a reference view $v_r$, we select M nearby views $\{v_{ni}\}_{i=i}^M$ along the camera trajectory. For each pixel $(u, v)$ in $v_r$, we sample a $3 \times 3$ patch $P_r(u, v)$ and warp it into the near views as $\{P'_{ni}(u, v)\}_{i=1}^M$ using Eq. 4. We then compute the averaged per-pixel variance among these patches as a reflection score for $v_r$, using standard deviation,

$$ref\,score = \frac{1}{9}\sum_{(x,y)\in P_r(u,v)} \text{std}(\Psi_r[P_r(u, v)], \Psi_{n1}[P'_{n1}(u, v)], ..., \Psi_{nM}[P'_{nM}(u, v)]), \tag{6}$$

where $\Psi$ denotes the normalized RGB image and $\text{std}(\cdot)$ is a per-pixel standard deviation operator. Since the reflected environments on the images may appear similar from certain viewing angles, the obtained per-view reflection scores are often inconsistent, as shown in Fig. 3 (a). To address this, we further introduce a spatial reflection fusion module to aggregate multi-view reflection scores. We back-project the per-view reflection scores into 3D space using depth maps to form a reflection score point cloud. For each query pixel, we perform a ball query [41] around its back-projected 3D location within the point cloud and compute the averaged top-$K$ scores, thus yielding the final reflection strength prior $w_{ref}$, as illustrated in Fig. 3 (b). The prior indicates how likely the surface has a high reflection strength, therefore can serve as a weight of the constraint on material maps $\Psi^M$,

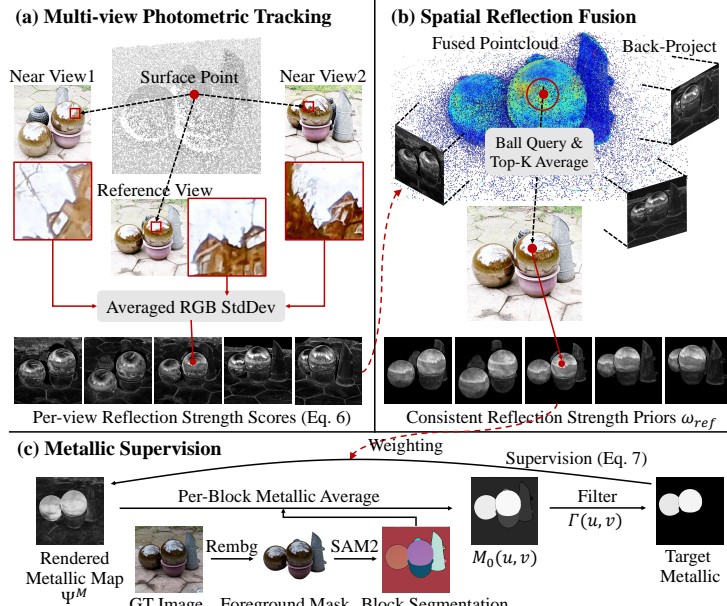

Figure 3: Illustration of computing reflection strength priors.

$$\mathcal{L}_{ref}(u,v) = \begin{cases} w_{ref} \cdot \Gamma(u,v) \cdot |M_0(u,v) - \Psi^M(u,v)| & \text{if } \Psi^M(u,v) < M_0 \\ 0 & \text{if } \Psi^M(u,v) \geq M_0 \end{cases}, \tag{7}$$

where $M_0(u,v)$ is a pre-computed target value, and $\Gamma(u,v)$ is a binary mask indicating whether the supervision is applied at pixel $(u,v)$. To adapt our constraint to centric scene structures, we extract foreground masks using [12] and segment them into semantically meaningful regions using SAM2 [43]. For each region, $\Gamma(u,v)$ is set to 1 if the averaged reflection prior $w_{ref}$ of all pixels in the region exceeds a threshold, and 0 otherwise. The target metallic value $M_0(u,v)$ is determined according to the averaged metallic value within each region. In a word, if a region more likely corresponds to a highly reflective surface, we apply a constraint that encourages higher metallic values for these pixels, as illustrated in Fig 3 (c). The intensity of the constraint is controlled by $w_{ref}$, allowing the supervision to be adaptively modulated based on our confidence of the surface reflectivity. Similar observations were discussed in [24, 56, 53], where highly reflective surfaces are often reliably identified early during the optimization.

**Normal Prior.** We also incorporate monocular normal priors obtained by a pre-trained network [57] to supervise the normals rendered by 2DGS. We find it to be an effective cue for scene geometry inference during the early training stage.

### 3.3 Environment Modeling through Ray Tracing

When using split-sum approximation to model the specular illumination in Eq. 3, an incident light $\omega_i$ fails to retrieve a plausible radiance from the environment map if it is occluded by other objects in the scene. To address this issue, we decompose the incident radiance into direct and indirect components, and introduce an occlusion probability $O(\omega_i) \in [0, 1]$ indicating how likely $\omega_i$ is occluded,

$$L_i(\omega_i) = L_{indirect}(\omega_i) + (1 - O(\omega_i)) \cdot L_{direct}(\omega_i). \tag{8}$$

The direct lighting $L_{direct}(\omega_i)$ can be obtained by querying a learnable environment map using the reflected direction and roughness. The environment map is a mip-mapped cubemap constructed with multiple roughness levels. To estimate the color of the indirect light which is occluded by other objects, we perform a Gaussian ray tracing [53]. Starting from a surface point, we trace a ray along the reflected direction and identify all intersected Gaussians, where each Gaussian has been transformed into a Bounding Volume Hierarchy (BVH) with two triangles. The intersected Gaussians

Table 1: Numrical evaluations on all four datasets. Best results are highlighted as 1st , 2nd , 3rd .

| Datasets | ShinyBlender [47] | | | | GlossySynthetic [33] | | | Ref-Real [47] | | | Mip-NeRF 360 [1] | | |
|---|---|---|---|---|---|---|---|---|---|---|---|---|---|
| Methods | PSNR↑ | SSIM↑ | LPIPS↓ | MAE°↓ | PSNR↑ | SSIM↑ | LPIPS↓ | PSNR↑ | SSIM↑ | LPIPS↓ | PSNR↑ | SSIM↑ | LPIPS↓ |
| RefNeRF [47] | 33.13 | 0.961 | 0.080 | 18.38 | 25.65 | 0.905 | 0.112 | 23.62 | 0.646 | 0.239 | - | - | - |
| ENVIDR [30] | 32.88 | 0.969 | 0.072 | 2.74 | 29.06 | 0.947 | 0.060 | 23.00 | 0.606 | 0.332 | - | - | - |
| 3DGS [22] | 30.37 | 0.947 | 0.083 | - | 26.01 | 0.886 | 0.089 | 23.85 | 0.660 | 0.230 | 27.21 | 0.815 | 0.214 |
| GaussianShader [20] | 31.94 | 0.957 | 0.068 | 7.00 | 27.11 | 0.922 | 0.082 | 23.46 | 0.521 | 0.257 | - | - | - |
| 2DGS [18] | 29.58 | 0.946 | 0.084 | - | 26.07 | 0.918 | 0.088 | 24.15 | 0.661 | 0.292 | 27.03 | 0.805 | 0.223 |
| 3DGS-DR [24] | 33.94 | 0.971 | 0.059 | 2.62 | 29.49 | 0.952 | 0.054 | 23.99 | 0.664 | 0.229 | 26.44 | 0.796 | 0.249 |
| Ref-Gaussian [56] | 35.04 | 0.973 | 0.056 | 4.59 | 30.08 | 0.957 | 0.050 | 24.61 | 0.685 | 0.252 | 26.62 | 0.781 | 0.272 |
| EnvGS [53] | 33.83 | 0.969 | 0.066 | 6.36 | 28.17 | 0.938 | 0.067 | 24.85 | 0.688 | 0.215 | 27.36 | 0.799 | 0.222 |
| Ours | 35.57 | 0.976 | 0.049 | 2.04 | 30.83 | 0.962 | 0.046 | 25.04 | 0.703 | 0.185 | 27.06 | 0.809 | 0.181 |

Figure 4: Visual comparisons on Synthetic Datasets. Our method successfully recovers fine-grained reflections on the helmet, as well as inter-reflection effects on the teapot.

are depth-sorted, and a splatting is performed to compute both the accumulated transmittance, denoted as $O(\omega_i)$, and the resulting lighting of the indirect illumination, denoted as $L_{indirect}(\omega_i)$,

$$L_{indirect}(\omega_i) = \sum_{i=1}^{N} c_d * o_i * p_i * \prod_{j=1}^{i-1}(1-o_j) + c_r, O(\omega_i) = \sum_{i=1}^{N} o_i * p_i * \prod_{j=1}^{i-1}(1-o_j), \quad (9)$$

where $N$ is the number of the intersected Gaussians during ray tracing. We also incorporate a residual term $c_r$ in the indirect radiance to account for noise and higher-order lighting effects [58, 56]. The ray tracing procedure naturally handles both occluded and unoccluded cases, where unoccluded rays yield $L_{indirect}(\omega_i) = O(\omega_i) = 0$. Therefore, we only need one ray tracing pass and one environment query to obtain the full incident radiance. Compared to Ref-Gaussian [56] which relies on an offline binary visibility indicator to separate direct and indirect terms and estimates indirect light solely through a residual color, our method evaluates occlusion in a fully differentiable manner. Moreover, this design allows Gaussians to participate in environment illumination modeling and be jointly optimized, leading to more physically grounded modeling and improved generalization.

### 3.4 Optimization

We train our method for a total of 30k iterations. We begin by training a 2DGS [18] with normal priors during the first 3k iterations to ensure geometric stability. After that, we incorporate PBR and our environment illumination modeling into the training process. At 10k iteration, we remove the normal prior to avoid potential bias from inaccurate predictions, and introduce our multi-view regularization terms. We also adopt normal propagation [24, 56] to propagate reliable normals to neighboring Gaussians for consistency and stability. The loss function can be written as

$$\mathcal{L} = \mathcal{L}_c + \lambda_{n-d}\mathcal{L}_{n-d} + \lambda_n\mathcal{L}_n + \lambda_{mv}\mathcal{L}_{mv} + \lambda_{ref}\mathcal{L}_{ref}, \quad (10)$$

where $\mathcal{L}_c = 0.8 * \mathcal{L}_{rgb} + 0.2 * \mathcal{L}_{D-SSIM}$ is the photometric loss commonly used in Gaussian-based methods [22, 56], $\mathcal{L}_{n-d}$ denotes the depth-normal consistency loss used in 2DGS [18], and $\mathcal{L}_n = |1 - N^T \hat{N}|$ is the normal prior loss. $\mathcal{L}_{mv}, \mathcal{L}_{ref}$ correspond to our multi-view consistency loss (Eq. 5) and reflection strength loss (Eq. 7), respectively.

## 4 Experiments

### 4.1 Experiment Settings

**Datasets & Metrics.** We evaluate the performance of our method on widely used benchmarks, including two synthetic datasets, ShinyBlender [47] and GlossySynthetic [33], as well as two real-

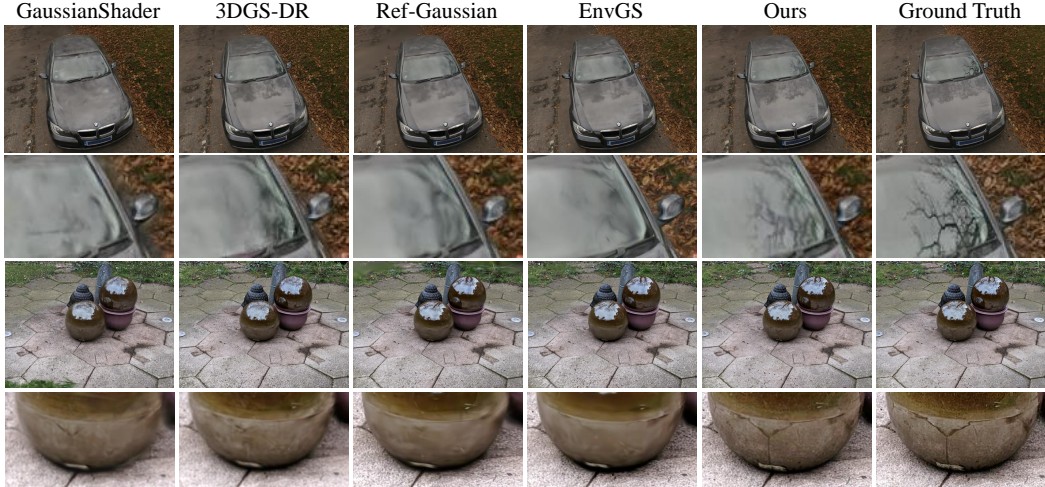

Figure 5: Visual comparisons in real-world Ref-Real [47] dataset. Our method accurately reconstructs reflection textures from surrounding environments on highly reflective surfaces.

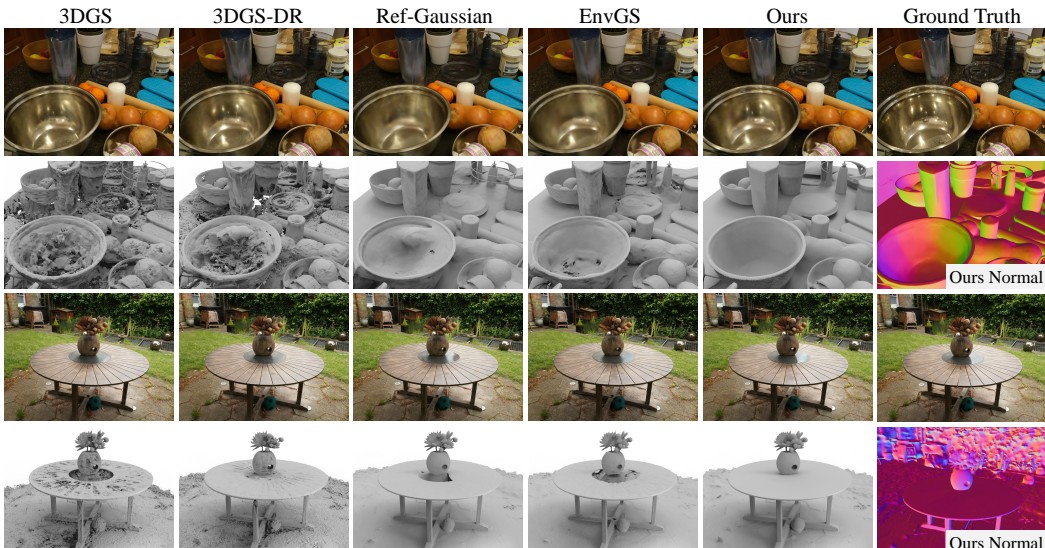

Figure 6: Visual comparisons in real-world Mip-NeRF 360 [1] dataset. Our method faithfully reconstructs highly reflective surfaces, such as the aluminum bowl and the metal plate.

world datasets, Ref-Real [47] and Mip-NeRF 360 [1]. All of these datasets contain challenging scenes with prominent reflective surfaces. To evaluate the quality of novel view synthesis, we report PSNR, SSIM [49] and LPIPS [59]. We also evaluate the accuracy of the predicted normals using Mean Angular Error (MAE).

**Baselines.** We compare our method with the state-of-the-art reflection modeling methods, including NeRF-based methods: Ref-NeRF [47], ENVIDR [30], as well as GS-based methods: 3DGS [22], GaussianShader [20], 2DGS [18], 3DGS-DR [24], Ref-Gaussian [56] and EnvGS [53].

## 4.2 Comparisons

**Comparisons on Synthetic Dataset.** We first evaluate our method on two synthetic datasets, ShinyBlender [47] and GlossySynthetic [33], and report the numerical results in Tab. 1, where we achieve the best performance across all metrics on both datasets. We further provide visual comparisons in Fig. 4, where our method accurately captures the environment reflections on the helmet. In addition, our approach effectively models secondary light reflections, such as the self-reflection of the teapot lid knob on the metallic lid, which benefits from our environment modeling.

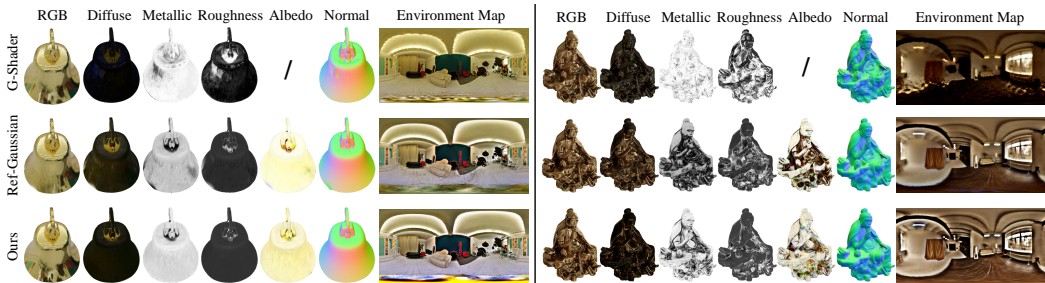

Figure 7: Visualization of illumination decomposition results and the learned environment maps. Our method produces more uniform and physically plausible material maps, along with sharper and more detailed environment maps. Best viewed with zoom in.

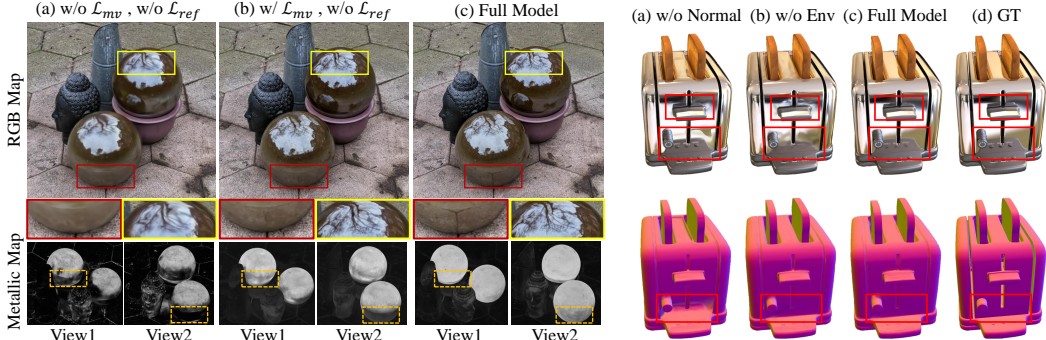

Figure 8: Ablation study on multi-view consistent material inference strategies.

Figure 9: Ablation study on normal prior and environment modeling.

**Comparisons on Ref-Real Dataset.** We also report quantitative comparisons on the real-world dataset Ref-Real [47] in Tab. 1, where our method consistently achieves state-of-the-art performance across all metrics. Visual comparisons in Fig. 5 show that our method accurately reconstructs the reflection textures from the surrounding environments on highly reflective surfaces, such as tree branches reflected on car windows and ground seams reflected on the metallic sphere.

**Comparisons on Mip-NeRF 360 Dataset.** We further evaluate our method on the more challenging real-world dataset Mip-NeRF 360 [1], as reported in Tab. 1. Existing reflective 3DGS methods often show degenerated performance on such complex environments with few reflective surfaces. In contrast, our method achieves competitive results and outperforms all baselines in terms of LPIPS, highlighting our strong generalization ability. Visual comparisons are provided in Fig. 6, where our method faithfully recovers highly reflective surfaces such as the aluminum bowl and the metal plate, which even remain difficult for existing GS-based reconstruction methods.

**Comparisons of Illumination Decomposition.** To validate the effectiveness of our illumination decomposition, we visualize the decomposed material components and the learned environment maps on GlossySynthetic dataset [33], as shown in Fig. 7. Note that our method cannot be directly compared with inverse rendering methods [11, 31, 21], as the illumination modeling approaches are much different.

## 4.3 Ablation Study

**Effectiveness of Each Module.** We conduct ablation studies to evaluate the effectiveness of each module in our framework on both synthetic and real-world datasets. We start by analyzing the multi-view material inference strategies. Without any strategies, the result (Fig. 8 (a), "w/o $\mathcal{L}_{mv}$, w/o $\mathcal{L}_{ref}$" row in Tab. 2) show inconsistent material maps and weak reflections. Introducing $\mathcal{L}_{mv}$ makes the multi-view material maps uniform and consistent, leading to clearer reflections on the top of the sphere (Fig. 8 (b), "w/ $\mathcal{L}_{mv}$, w/o $\mathcal{L}_{ref}$" row in Tab. 2). Further adding $\mathcal{L}_{ref}$ improves the metallic in textureless reflective regions, making subtle reflections like ground seams more visible (Fig. 8 (c), "Full Model" row in Tab. 2). We also ablate the environment modeling strategy by removing ray tracing, relying on residual color and environment map to overfit incident illumination. This causes

Table 2: Ablation study on each one of our modules.

| Datasets | ShinyBlender [47] | | | Ref-Real [47] | | |
|---|---|---|---|---|---|---|
| Models | PSNR↑ | SSIM↑ | LPIPS↓ | PSNR↑ | SSIM↑ | LPIPS↓ |
| w/o $\mathcal{L}_{mv}$, w/o $\mathcal{L}_{ref}$ | 34.87 | 0.972 | 0.055 | 24.24 | 0.655 | 0.260 |
| w/ $\mathcal{L}_{mv}$, w/o $\mathcal{L}_{ref}$ | 35.21 | 0.975 | 0.051 | 24.47 | 0.670 | 0.242 |
| w/o $\mathcal{L}_n$ | 35.37 | 0.975 | 0.050 | 24.39 | 0.672 | 0.229 |
| w/o Environment | 34.69 | 0.976 | 0.049 | 24.76 | 0.681 | 0.199 |
| Full Model | **35.57** | **0.976** | **0.049** | **25.04** | **0.703** | **0.185** |

noticeably blurred reflections in inter-reflection regions (Fig. 9 (b), "w/o Environment" row in Tab. 2). Lastly, removing the normal prior from full model leas to degenerated geometry and color (Fig. 9 (a), "w/o $\mathcal{L}_n$" row in Tab. 2).

**Normal Prior.** To evaluate the necessity of the normal prior, we conduct ablation studies on ShinyBlender dataset [47] under three experimental settings: Full Model, Full model without normal prior (w/o $\mathcal{L}_n$), Full model without normal prior, material regularization and environment modeling (w/o $\mathcal{L}_n$, w/o Reg, w/o Env). We report the normal accuracy in Tab. 3 using

Table 3: Ablation study on normal prior.

| Models | MAE↓ | CD↓ |
|---|---|---|
| w/o $\mathcal{L}_n$, w/o Reg, w/o Env | 3.47 | 0.94 |
| w/o $\mathcal{L}_n$ | 2.59 | 0.68 |
| Full Model | 2.04 | 0.60 |

MAE, as well as the geometric reconstruction accuracy compared with ground truth meshes using Chamfer Distance (CD). The results indicate that, beyond the normal prior, our material regularization and environment modeling also contribute significantly to geometry learning. This is because both our material constraints and environment modeling are differentiable to the depth and normal, enabling end-to-end joint optimization of geometry, appearance, and material properties for improved overall performance.

# 5 Conclusion

We propose MaterialRefGS, a novel approach that learns multi-view illumination decomposition for reflective gaussian splatting through multi-view consistent material inference. To this end, we enforce the Gaussians to produce consistent material maps across different views, and explore reflection strength priors from photometric variants to provide explicit supervision for specular reflectance modeling. We also introduce a novel environment modeling strategy based on Gaussian ray tracing, which compensates for the indirect illumination caused by inter-object occlusion. Extensive ablation studies justify the effectiveness of our proposed modules, loss functions, and training strategies. Our evaluations show our superiority over the latest methods in rendering photorealistic novel views and recovering accurate geometry.

# 6 Acknowledgement

This work was supported by Deep Earth Probe and Mineral Resources Exploration – National Science and Technology Major Project (2024ZD1003405), and the National Natural Science Foundation of China (62272263), and in part by Kuaishou.

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
