# OpenReview forum: "MaterialRefGS: Reflective Gaussian Splatting with Multi-view Consistent Material Inference"
_NeurIPS.cc/2025/Conference — NeurIPS 2025 poster_

### Official Review · Reviewer_dUyL · 2025-06-28

**Clarity:** 3
**Significance:** 4
**Originality:** 3
**Rating:** 5
**Confidence:** 5

**Summary:**

This paper proposes a multi-view consistent framework for learning reflections in 3D Gaussian Splatting (3DGS), enhancing material inference and environment modeling. By enforcing consistent material maps and leveraging ray-traced indirect illumination, the method improves photorealism and achieves state-of-the-art novel view synthesis.

**Questions:**

Kindly refer to the [Weaknesses].

**Ethical Concerns:**

["NO or VERY MINOR ethics concerns only"]

**Final Justification:**

The authors' response and additional experiments have addressed my concerns. Considering the perspectives of the other reviewers, I will maintain my current score.

**Limitations:**

Yes

**Quality:**

4

**Strengths And Weaknesses:**

### Strengths
* The motivation of supervising the learning of accurate reflective surfaces through multi-view material attribute consistency is well-founded, and the effectiveness of the proposed method is thoroughly validated through experiments.
* Leveraging visual foundation models to provide additional prior input proves highly suitable for such ill-posed tasks, and the authors present sufficiently convincing visual results.
* The paper is well-written and easy to follow.

### Weaknesses
*  There is some confusion regarding the simultaneous learning of albedo and diffuse attributes. The simplified Disney BRDF model employed in the paper treats albedo and diffuse components as independent, which is uncommon in computer graphics literature. Why not compute diffuse from albedo, rather than optimizing both separately? In L161–162, the authors claim that the contribution of “albedo” to non-reflective surfaces is minimal, which contradicts the traditional definition of albedo in graphics. If the authors could provide experiments to demonstrate that separating albedo and diffuse terms significantly benefits the optimization of 3DGS, it would help dispel this confusion.
* The choice of values for $M_0$ and $\Gamma$ in Equation 7 seems rather ad hoc. Whether Eq. 7 functions effectively depends heavily on the specific values of $M_0$  and $\Gamma$ yet the paper lacks experimental validation to confirm the robustness of their computation. It is recommended to include ablation studies on the thresholding of the averaged reflection prior $w_{ref}$, as well as on the computation method for $M_0$ .
* Visualization of occlusion and indirect illumination. Although the authors provide qualitative and quantitative comparisons with and without environment modeling, many recent works employ similar techniques involving ray tracing or residual color estimation for indirect illumination. While I believe the proposed one-bounce ray tracing with residual color is a promising solution, I am curious about how the resulting occlusion and indirect illumination buffers compare with those of other methods such as GS-IR, Ref-Gaussian, TensoIR, or Relightable 3DGS.
* The normal prior. As illustrated in the visualizations in Fig. 9, the normal prior appears to be critical, particularly for reconstructing geometric surfaces. However, do the two regularization strategies proposed by the authors rely on the accuracy of the geometric surface in the initial optimization stage? If the monocular normal prediction model provides inaccurate or viewpoint-inconsistent prior sets, can the proposed method still serve as a robust regularizer for material learning? Additionally, in Tab. 2, two rows of metrics for ShinyBlender are identical, which may be a minor oversight.

---

> ### Author Rebuttal · Authors · 2025-07-30
>
> Dear reviewer, thanks for your valuable comments and recognizing our **_well-founded motivation_** and **_well-organized structures_**. We are glad to provide further clarifications and experiments to address your concerns.
>
> **1. About albedo term.**
>
> The primary goal of our work is modeling reflection effects in complex scenes for novel view synthesis. Therefore, following Ref-Gaussian, we directly rasterize the diffuse color $c_{diffuse}$ using per-Gaussian spherical harmonics and use other material properties such as albedo, metallic, and roughness to evaluate the specular color $c_{specular}$.
> We observe that incorporating albedo in modeling the diffuse component, as done in traditional graphics, will significantly increase the optimization difficulty, especially in complex real-world scenes. On the other hand, since the diffuse component is not sensitive to viewing direction, delegating the prediction of diffuse color to the Gaussians allows us to better disentangle reflection effects from illumination. The full rendering equations we used can be written as follows,
>
> $$
> \begin{aligned}
> c_{\text{diffuse}} &= (1 - m)c_d, \\\\
> c_{\text{specular}} &= \left(((1 - m) \cdot 0.04 + m \cdot a) \cdot F_1 + F_2 \right) \cdot \text{env}(\omega_i, r), \\\\
> F_1,\ F_2 &= \text{lookuptable}(\omega_i, r), \\\\
> c_{\text{final}} &= c_{\text{diffuse}} + c_{\text{specular}}
> \end{aligned}
> $$
>
> where $ \text{env}(\omega_i, r) $ denotes querying from the mip-mapped environment cube maps using the incident direction and roughness. For indirect illumination, this term is replaced by a combination of the residual color and the color obtained via 2D Gaussian ray tracing. $ F_1, F_2 $ are obtained through table lookup based on the incident direction and roughness. From the above equation, we can see that the albedo component contributes little to the specular color when the metallic value is small. That is why we claim that the contribution of albedo to non-reflective surfaces is minimal. Through this way, we do not need to devote much attention to learning albedo in non-reflective regions, which is why we argue that our modeling strategy significantly reduces the optimization difficulty of material decomposition.
>
> To further demonstrate this, we compare our method with inverse rendering methods which use albedo to compute both diffuse and specular components on the real-world Mip-NeRF 360 dataset. As reported in the table below, our superior results highlight the advantages of our decomposition approach in complex real-world scenes.
>
>
> | Methods            | PSNR$\uparrow$ | SSIM$\uparrow$ | LPIPS$\downarrow$ |
> |--------------------|----------------|----------------|-------------------|
> | GS-IR | 25.38          | 0.757          | 0.267             |
> | GS-ID   | 25.91          | 0.771          | 0.264             |
> | IRGS     | 21.93          | 0.525          | 0.428             |
> | Ours                 | **27.06**      | **0.809**      | **0.181**         |
>
> **2. The choice of $M_0$ and $\Gamma$.**
>
> In our implementation, $\Gamma$ is set to 1 or 0 according to whether the average metallic value in a region is larger than 0.5 or not, practically. And $M_0$ is set to be the top 30\% of metallic values in a region, practically. We conduct additional ablation studies on these two thresholds, as reported below.
>
> | | $M_0$      | $M_0$      | $M_0$      | $M_0$      | $M_0$      | $\Gamma$    | $\Gamma$    | $\Gamma$    | $\Gamma$    | $\Gamma$    |
> |-------------------|------------|------------|------------|------------|------------|-------------|-------------|-------------|-------------|-------------|
> |   Metric    | 10\%       | 30\%       | 50\%       | 70\%       | 90\%       | 0.1         | 0.3         | 0.5         | 0.7         | 0.9         |
> | **PSNR$\uparrow$** | 23.435     | **23.438** | 23.433     | 23.426     | 23.414     | 23.288      | **23.444**  | 23.438      | 23.437      | 23.353      |
> | **SSIM$\uparrow$** | 0.6436     | **0.6439** | 0.6434     | 0.6427     | 0.6412     | 0.6329      | 0.6438      | **0.6439**  | **0.6439**  | 0.6378      |
> | **LPIPS$\downarrow$** | 0.1871  | **0.1870** | 0.1875     | 0.1891     | 0.1897     | 0.1943      | 0.1874      | **0.1870**  | 0.1873      | 0.1917      |
>
> For $M_0$, too small threshold makes the reflection strength prior ineffective, while too large threshold results in surfaces with overly strong reflection strength, both of which harm the modeling of accurate reflection effects. For $\Gamma$, too small value causes most regions to be wrongly identified as highly reflective surfaces, while too large value fails to identify true surfaces with high reflectiveness, both of which hinders the learning of material decomposition. Overall, however, our method remains robust to the choice of these thresholds.
>
> **3. Visualization of occlusion and indirect illuminance.**
>
> Due to the policy restriction, we are unable to show visualization of the occlusion maps or indirect illumination. To quantitatively evaluate the results, we compute ground truth binary indirect illumination masks and occlusion masks for each view on the ground truth meshes. We then evaluate the rendering quality in indirect regions and the accuracy of binarized occlusion maps on ShinyBlender and GlossySynthetic datasets. We compare our method with both reflection modeling method Ref-Gaussian and inverse rendering methods including TensoIR, Relightable3DGS, GS-IR. For fair comparison, we add monocular normal priors for all baseline methods.
>
> The results reported in the table below significantly show our advantages in approximating indirect illumination and localizing occlusions. The reason is that our multi-view material inference and environment modeling are end-to-end differentiable with respect to geometry, which allows for joint optimization of geometry, appearance, and material decomposition. In contrast, baseline methods typically first optimize geometry and then use the fixed geometry to predict occlusion and perform inverse rendering, which makes them less flexible in handling complex regions.Additionally, the material modeling strategy adopted in inverse rendering methods is not well-suited for highly reflective scenes, as we analyzed in our response to the first question "About the albedo term". We will add more visualization results in terms of indirect illumination and occlusion maps in the revised version.
>
>
> | | Indirect | Illumination  //// | Occlusion | Maps   |
> |--------------------|-------------------------------------|--------------------------------------|----------------------------------|----------------------------------|
> | **Methods**            | **PSNR$\uparrow$** |  **SSIM$\uparrow$** |  **Accuracy$\uparrow$** |  **F1-score$\uparrow$** |
> | TensoIR            | 18.63                               | 0.828                                | 0.7514                           | 0.3944                           |
> | Relightable3DGS    | 17.26                               | 0.827                                | 0.7409                           | 0.3480                           |
> | GS-IR              | 15.06                               | 0.754                                | 0.7153                           | 0.3065                           |
> | Ref-Gaussian       | 24.60                               | 0.907                                | 0.9133                           | 0.8279                           |
> | Ours               | **27.91**                           | **0.951**                            | **0.9492**                       | **0.9244**                       |
>
> **4. About numerical results in Tab. 2.**
>
> Sorry for the mistake. We wrongly reported the third row (w/o $\mathcal{L}_n$) of ShinyBlender dataset in Tab. 2. It should be 35.37 / 0.975 / 0.050. We will correct this in the revised version.
>
> **5. About normal prior.**
>
> The normal prior is helpful for better convergence of geometry in the early training stage, which indeed benefits subsequent multi-view material inference and environment modeling. However, our method does not heavily rely on the normal prior. As shown in the third row of Tab. 2 of the main paper, even without the normal prior, our method still achieves state-of-the-art results on the ShinyBlender dataset. In fact, even without perfect geometry, our multi-view material inference and environment modeling continue to improve the geometry in later training stages. This is because our material constraints and environment modeling are differentiable to the depth and normal, enabling end-to-end joint optimization of geometry, appearance, and material properties for improved overall performance.
>
> We further change different monocular normal estimation models and report our rendering results with different normal priors, as reported in the table below. We achieve similar outstanding results based on various normal priors, which demonstrates our robustness to imperfect normal priors.
>
> | Methods                 | PSNR$\uparrow$ | SSIM$\uparrow$ | LPIPS$\downarrow$ |
> |-------------------------|----------------|----------------|-------------------|
> | GeoWizard[1]      | 23.37          | 0.638          | 0.193             |
> | StableNormal[2] | 23.41          | 0.640          | 0.189             |
> | Metric3D-v2[3]   | **23.44**      | **0.644**      | **0.187**         |
>
> [1]. Xiao Fu, Wei Yin, et al. Geowizard: Unleashing the diffusion priors for 3d geometry estimation from a single image. In European Conference on Computer Vision, pages 241–258. Springer, 2025.
>
> [2]. Chongjie Ye, Lingteng Qiu, et al. StableNormal: Reducing diffusion variance for stable andsharp normal. ACM Transactions on Graphics (TOG), 43(6):1–18, 2024.
>
> [3]. Mu Hu, Wei Yin, Chi Zhang, et al. Metric3d v2: A versatile monocular geometric foundationmodel for zero-shot metric depth and surface normal estimation. IEEE Transactions on PatternAnalysis and Machine Intelligence, 2024.

---

> > ### Author Response · Authors · 2025-08-06
> >
> > Dear Reviewer,
> >
> > We sincerely appreciate your insightful and valuable comments. We hope that our rebuttal has addressed your concerns. With the discussion period deadline approaching, if you have any remaining questions, please do not hesitate to let us know. We highly value this opportunity for discussion and would be happy to provide additional details or explanations.
> >
> > Best,
> >
> > The authors

---

> > > ### Comment · Reviewer_dUyL · 2025-08-06
> > >
> > > Thank you for the rebuttal — it is rigorous, detailed, and thoroughly addresses my concerns. Excellent work. I encourage the authors to include the additional experiments and visualization results in the final version, and I look forward to the public release of the code.

---

> > > > ### Author Response · Authors · 2025-08-07
> > > >
> > > > Dear Reviewer,
> > > >
> > > > We sincerely appreciate your recognition and accept recommendation of our work. We will include the additional experiments and visualization results in the final version. The code will be released publicly upon acceptance.
> > > >
> > > > Best regards,
> > > >
> > > > The Authors

---

### Official Review · Reviewer_Th9t · 2025-06-30

**Clarity:** 1
**Significance:** 2
**Originality:** 2
**Rating:** 4
**Confidence:** 5

**Summary:**

This paper introduces MaterialRefGS, a novel approach to model reflection with multi-view material consistency and reflection strength prior supervision. Inspired by multi-view stereo methods, material maps are constrained to be view-independent by avoiding view-specific overfitting. To facilitate this process,  reflection scores are evaluated to supervise reflection strength attributes such as metallic parameter. The proposed method also proposes a differentiable environment model for photorealistic rendering with indirect illumination.  MaterialRefGS achieves state-of-the-art results of novel view synthesis for various datasets.

**Questions:**

1. Why does the reflection strength prior regularize the metallic parameter rather than the roughness? Roughness has a more direct impact on surface reflectivity.
2. The improvement in geometry reconstruction seems to stem from the monocular normal prior rather than from reflection or environment modeling. Please provide further ablation studies isolating the effect of the normal prior, including MAE scores and normal visualization.
3. The assumption that the same semantic region shares similar material properties is questionable. The SAM2 model is not trained to distinguish between different material properties.
4. In the presence of self-occlusion shadows, is it possible to separate shadows from intrinsic material properties? Shadows may be baked into the diffuse map, complicating reflectance decomposition.
5. The distinction between "diffuse" and "albedo" in the paper is unclear. "Diffuse" refers to diffuse albedo, while "albedo" appears to denote specular albedo. However, in the commonly used Disney BRDF model, both diffuse and specular components share the same albedo parameter.

I hope authors address my concerns in Weaknesses and Questions to increase my score.

**Ethical Concerns:**

["NO or VERY MINOR ethics concerns only"]

**Final Justification:**

The rebuttal provided detailed explanations of the proposed design and implementation, addressing most of my initial concerns.

The additional experiments were thorough and effectively demonstrated the method’s effectiveness.

I acknowledge the significance and potential impact of the environment modeling component and multi-view material consistency.

Overall, I raise my score to Borderline Accept, and I encourage inclusion of more comprehensive visualizations, explanations, and release of public code.

**Limitations:**

yes

**Paper Formatting Concerns:**

No concerns

**Quality:**

2

**Strengths And Weaknesses:**

Strengths
1. Considering multi-view material consistency of view-independent material maps significantly improves performance.
2. The reflection strength prior is effectively used to encourage higher metallic values for pixels with high weights.
3. MaterialRefGS achieves SOTA performance in novel view synthesis

Weaknesses
1. Additional visualizations of the reflection strength prior across diverse scenes are necessary to demonstrate that highly reflective surfaces consistently exhibit higher weights.
2. More visualizations of indirect illumination for glossy objects are needed to validate the effectiveness of the proposed environment modeling. Indirect radiance should handle the inter-reflection of the surfaces.
3. The proposed differentiable environment modelling is an extension of prior methods, combining 2D Gaussian ray tracing [1] and Ref-Gaussian [2].

References

[1] IRGS: Inter-Reflective Gaussian Splatting with 2D Gaussian Ray Tracing, CVPR 2025

[2] Reflective Gaussian Splatting, ICLR 2025

---

> ### Author Rebuttal · Authors · 2025-07-30
>
> Dear reviewer, thanks for your valuable comments and suggestions. We are glad to provide further clarifications and experiments to address your concerns.
>
> **1. Visualization of reflection strength prior.**
>
> Due to the policy restriction, we are unable to show visualization of the reflection strength priors. To quantitatively evaluate the effectiveness, we binarize our reflection strength prior and uses the ground truth materials to identify highly reflective surface regions that are either predicted or the ground truth. We then compute the Accuracy and F1-score of the classification results on GT meshes between the predicted and ground truth regions, as reported below. The experiments are conducted on ShinyBlender dataset where there are ground truth meshes and materials. The results demonstrate that our reflection strength prior accurately localizes the highly reflective surfaces. We will include more visualization of the reflection strength prior in the revised version.
>
> | |ball|car|toaster|helmet|coffee|teapot|Mean|
> |-|-|-|-|-|-|-|-|
> |**Accuracy↑**|98.82%|96.79%|96.10%|95.73%|96.30%|94.69%|96.40%|
> |**F1-score↑**|0.9817|0.9564|0.9612|0.9545|0.9557|0.9459|0.9592|
>
> **2. Visualization of indirect illumination.**
>
> We have included several examples of indirect illumination in the main paper, such as:
> - the reflection of the flowerpot in the tray at the bottom in the first row of Fig.1;
> - the reflection of the teapot lid’s handle on the lid surface in the second row of Fig.4;
> - the reflection of ground seams on the metallic ball in the fourth row of Fig.5;
> - inter-reflections in the first and third rows of Fig.2 in the supplementary material.
>
> Unfortunately, due to the policy restriction, we are unable to provide more visualization here. As an alternative, we evaluate the rendering performance in indirect illumination regions labeled by ground truth masks calculated on the ground truth meshes from each view. The experiments are conducted on ShinyBlender and GlossySynthetic datasets, where there are ground truth meshes. The results are reported in the table below. We do not include LPIPS metric here, as it is designed for full image evaluation. Our method shows significant improvements over the baselines in indirect illumination regions, demonstrating the effectiveness of our environment modeling module. We will include more visual examples in the revised version.
>
> |**Scenes**|**toaster**| |**coffee**| |**teapot**| |**tbell**| |**angel**| |
> |-|-|-|-|-|-|-|-|-|-|-|
> |**Methods**|**PSNR↑**|**SSIM↑**|**PSNR↑**|**SSIM↑**|**PSNR↑**|**SSIM↑**|**PSNR↑**|**SSIM↑**|**PSNR↑**|**SSIM↑**|
> |GaussianShader|18.24|0.821|32.70|0.951|12.67|0.723|22.02|0.951|17.83|0.876|
> |3DGS-DR|18.93|0.839|33.55|0.953|14.81|0.792|26.26|0.968|18.96|0.901|
> |Ref-Gaussian|20.60|0.852|34.72|0.955|17.51|0.835|28.46|0.970|21.71|0.921|
> |EnvGS|19.20|0.844|33.17|0.954|13.04|0.776|23.94|0.962|18.17|0.892|
> |**Ours**|**25.97**|**0.942**|**35.37**|**0.959**|**21.50**|**0.912**|**31.14**|**0.985**|**25.56**|**0.957**|
>
> **3. About environment modeling module.**
>
> Ref-Gaussian separates surfaces into direct and indirect illumination regions and uses rendered residual colors from Gaussians to approximate indirect radiance. However, this approach has significant limitations in modeling complex inter-reflections between objects. IRGS introduces 2D Gaussian ray tracing, but it requires Monte Carlo sampling at every ray-surface intersection to evaluate the rendering equation, which significantly slows down the training and inference, and makes it less scalable to complex real-world scenes.
>
> Our method combines the strengths of both approaches. We apply the split-sum approximation in indirect illumination regions and conduct 2D Gaussian ray tracing along the reflected rays to model higher-order lighting effects. As a result, we achieves significantly better rendering quality. Qualitative comparisons can be found in the second row of Fig. 4, the fourth row of Fig. 5 in the main paper, and the first, third, and fourth rows of Fig. 2 in the supplementary material. The numerical comparisons in the indirect illumination regions, as shown in the table above, further highlight the strengths of our approach.
>
> **4. Why not supervise roughness & The use of albedo in diffuse and specular color.**
>
> The primary goal of our work is modeling reflection effects in complex scenes for novel view synthesis. Therefore, our approach of material modeling and its usage differs from inverse rendering methods such as IRGS and GS-IR.  Following Ref-Gaussian, we directly rasterize the diffuse color $c_{diffuse}$ using per-Gaussian spherical harmonics and use other material properties such as albedo, metallic, and roughness to evaluate the specular color $c_{specular}$, rather than using albedo and metallic to compute both diffuse and specular components as done in inverse rendering methods. This design avoids the challenging task of evaluating albedo and metallic values for all locations in complex real-world scenes. Moreover, since the diffuse component is not sensitive to viewing direction, delegating the prediction of diffuse color to the Gaussians allows us to better disentangle reflection effects from illumination. In such case, our rendering equation can be specifically written as follows,
>
> $$
> \begin{aligned}
> c_{\text{diffuse}}&=(1-m)c_d,\\\\
> c_{\text{specular}}&=\left(((1-m)\cdot 0.04+m\cdot a)\cdot F_1+F_2\right)\cdot\text{env}(\omega_i, r),\\\\
> F_1,\ F_2&=\text{lookuptable}(\omega_i, r),\\\\
> c_{\text{final}}&=c_{\text{diffuse}}+c_{\text{specular}}
> \end{aligned}
> $$
>
> Where $\text{env}(\omega_i,r)$ denotes querying from the mip-mapped environment cube maps using the incident direction and roughness. For indirect illumination, this term is replaced by a combination of the residual color and the color obtained via 2D Gaussian ray tracing. $F_1,F_2$ are obtained through table lookup based on the incident direction and roughness.
>
> From the above equation, we can see that roughness only influences our color modeling indirectly, whereas the metallic value directly determines the proportion of light attributed to specular and diffuse components. Therefore, we apply the reflection strength prior as a regularization term for the metallic prediction.
>
> **5. Ablation on normal prior.**
>
> We conduct additional ablation studies on ShinyBlender dataset under three experimental settings:  **A. Full model**;  **B. Full model without normal prior**; **C. Full model without normal prior and without other components (including material regularization and environment modeling)**.  We report the normal accuracy using MAE, as well as the geometric reconstruction accuracy using Chamfer Distance (CD) below.
>
> | Metric|A|B|C|
> |-|-|-|-|
> | MAE$\downarrow$|**2.04**|2.59|3.47|
> | CD$\downarrow$|**0.60**|0.68|0.94|
>
> The results indicate that, beyond the normal prior, our material regularization and environment modeling also contribute significantly to geometry learning. This is because both our material constraints and environment modeling are differentiable to the depth and normal, enabling end-to-end joint optimization of geometry, appearance, and material properties for improved overall performance.
>
> **6. About the segmentation of SAM2.**
>
> Sorry for the misunderstanding in Fig.3 and description. We don't enforce SAM2 to perform semantic segmentation of the scene. Instead, we encourage SAM2 to distinguish patterns that exhibit different appearances due to varying material properties, which is not limited to semantic objects. It can divide a single object into multiple patches to identify regions with different material attributes. Furthermore, the segmentation granularity of SAM2 is controllable by configuring the number of anchor points per side and specifying whether to prioritize traversal based on minimum or maximum patch areas. By setting an appropriate granularity level, we can prevent regions with different materials from being merged into a single part. We provide an additional ablation study on the impact of segmentation granularity, and report the granularity level, the average number of patches predicted by SAM2, and the rendering metrics, as shown in the table below.
>
> |Granularity|PointsPerSide|Patch Traversal|Number of Patches|PSNR$\uparrow$|SSIM$\uparrow$|LPIPS$\downarrow$|
> |--|--|--|--|--|--|--|
> |G1|128|From Min to Max|15|**30.63**|**0.961**|**0.046**|
> |G2|64|From Min to Max|9.5|30.54|0.961|0.047|
> |G3|64|From Max to Min|2.5|30.09|0.958|0.050|
> |G4|32|From Max to Min|1|29.85|0.957|0.052|
>
> In the table, G1 represents the finest segmentation granularity, while G4 corresponds to the coarsest. The experimental results show that finer granularity leads to better performance, however, the difference is not significant. Our method exhibits robustness to the choice of segmentation granularity. We use the same setting as G1 in our implementation. We will add more segmentation visualizations in the revised version.
>
> **7. About self-occlusion shadows.**
>
> Our method is able to effectively handle inter-reflection effects caused by self-occlusion between reflective objects. As for the shadow cast by one object onto another behind it, it may indeed be learned in the diffuse component. However, our method mainly focuses on modeling reflection effects in the scene for novel view synthesis. Since the light sources in the scene are fixed, shadows can be naturally interpreted as an intrinsic component of the diffuse color. Therefore, the shadows in the diffuse map do not harm the rendering quality from different views. We will add visualizations of multi-view rendered shadows in the revised version to demonstrate this. To explicitly disentangle shadows from illumination, future works may include incorporating image relighting priors to remove shadows from input images, or leveraging additional input images captured under varying lighting conditions to provide richer illumination priors.

---

> ### Comment · Reviewer_Th9t · 2025-08-04
> **Response for author's rebuttal**
>
> I appreciate the authors' thorough and thoughtful rebuttal. Most of my concerns regarding the weaknesses and questions have been adequately addressed.
> The additional experiments are detailed and clearly demonstrate the effectiveness of the proposed method.
>
> However, the contribution related to environment map modelling still feels limited, as it primarily combines the strengths of existing methods without introducing substantial novelty.
>
> Based on the overall improvements, I have decided to raise my score to Borderline Reject.
>
> I hope that more comprehensive visualizations will be included in the final version of the paper.

---

> > ### Author Response · Authors · 2025-08-05
> > **Response for reviewer**
> >
> > We are very pleased to know that our rebuttal addressed most of your concerns and that you raised your rating. We will include more comprehensive visualizations in the final version. However, we would like to provide further clarification regarding the environment modeling module.
> >
> > **First, we would like to clarify that our core contribution lies in exploring multi-view consistent knowledge to guide and supervise the material inference process.** To the best of our knowledge, this perspective has not been explored in previous works on either reflection modeling or inverse rendering.
> >
> > Building upon our core contribution, we additionally introduce 2D Gaussian ray tracing into our environment modeling. While we acknowledge that this is not a groundbreaking innovation, it undoubtedly serves as a valuable complement to our core contribution and effectively mitigates the common limitations of existing reflection modeling methods.
> >
> > **Second, we want to emphasize that our environment modeling strategy is not a trivial combination of Ref-Gaussian and 2D Gaussian ray tracing.** The primary differences is: The visibility map of Ref-Gaussian is computed offline and therefore suffers from inaccuracies due to imperfect geometry. Instead, we discard the offline mesh and dynamically compute the incident light from the rendered depths and normals. We then perform ray tracing along the incident direction to obtain an alpha map and color map. Then we blend the color map with the queried environment map along the same direction, using the alpha map as weights. In this way, the ray tracing process is seamlessly integrated with our visibility and incident light direction computation in a differentiable manner, allowing indirect illumination and scene geometry to be jointly optimized in a flexible and end-to-end fashion.
> >
> > We report a straightforward modification that directly combines Ref-Gaussian and 2D Gaussian ray tracing, as shown in the third row of the two tables below, which report the rendering quality in indirect regions. Our experiments show that such a naive combination does not lead to significant performance improvements.
> >
> > | ShinyBlender  | Indirect Regions |  |
> > |----------------|----------|---------|
> > | **Methods**        | **PSNR** ↑   | **SSIM** ↑  |
> > | IRGS           | 16.62    | 0.765   |
> > | Ref-Gaussian   | 27.66    | 0.904   |
> > | Ours (Naive)   | 28.82    | 0.911   |
> > | **Ours**       | **30.67** | **0.951** |
> >
> > | GlossySynthetic | Indirect Regions  |  |
> > |----------------|----------|---------|
> > | **Methods**       | **PSNR** ↑   | **SSIM** ↑  |
> > | IRGS           | 13.78    | 0.770   |
> > | Ref-Gaussian   | 22.56    | 0.909   |
> > | Ours (Naive)   | 23.47    | 0.914   |
> > | **Ours**       | **26.07** | **0.952** |
> >
> > It is worth noting that we also attempted to modify Ref-Gaussian to support dynamic visibility map, but observed significant degeneration of the geometry. Further analysis revealed that the inclusion of ray tracing introduced additional challenges for geometry optimization. Interestingly, this issue was resolved after incorporating the multi-view material consistency constraint. This is because the constraint plays a critical role in preserving and promoting the structural integrity of the scene geometry.
> >
> > We apologize that we should have provided a more thorough discussion of this module in the appendix. In the revised version, we will include a detailed explanation of our design and implementation. We hope these clarifications address your concerns regarding the environment modeling module. If you have any further considerations, please do not hesitate to let us know, as we would be glad to engage in further discussion.

---

> > > ### Comment · Reviewer_Th9t · 2025-08-05
> > > **Response for reviewer**
> > >
> > > I appreciate the detailed explanation of the proposed design and implementation.
> > >
> > > I fully recognize the significance of the environment modeling and agree with its potential impact.
> > >
> > > Based on these clarifications, I have decided to raise my score to Borderline Accept.
> > >
> > > I hope that the final version of the paper and supplementary materials will include more comprehensive visualizations and explanations, and that the authors will make the code publicly available in the future.

---

> > > > ### Author Response · Authors · 2025-08-05
> > > > **Response for reviewer**
> > > >
> > > > Dear Reviewer,
> > > >
> > > > We sincerely appreciate your recognition of our clarifications and explanations regarding the environment modeling module, and we are truly grateful for your accept recommendation. We will definitely include more visualizations and detailed explanations of the issues discussed in the rebuttal in the final version and supplementary materials. Our code will be made publicly available upon acceptance of the paper.
> > > >
> > > > Best,
> > > >
> > > > The authors

---

### Official Review · Reviewer_t1W5 · 2025-07-01

**Clarity:** 3
**Significance:** 3
**Originality:** 3
**Rating:** 5
**Confidence:** 5

**Summary:**

This paper addresses the challenge of handling reflective scenes using 3DGS. The authors argue that multi-view consistent material inference is crucial for the task. The proposed approach enforces Gaussian Splatting to generate view-consistent material maps while utilizing photometric variations across multiple views as prior information for reflective regions. Experiments on public benchmarks demonstrate that the proposed method achieves SOTA performance in novel view synthesis.

**Questions:**

The authors could also refer to the weakness section:

- Better visualizations and explanations for Equation 1 and Figure 3 are suggested.
- The baselines used are a bit confusing.
- Ablation studies on window size parameters are suggested.

I would consider raise my rating if the above concerns could be addressed.

**Ethical Concerns:**

["NO or VERY MINOR ethics concerns only"]

**Final Justification:**

The authors have addressed my concerns in the rebuttal, and I would raise my final rating to "accept".

I think the paper would make a good NeurIPS poster paper.

**Limitations:**

Yes

**Paper Formatting Concerns:**

-

**Quality:**

3

**Strengths And Weaknesses:**

Strengths:
- The proposed method performs well in real-world scenes, with high-quality mesh results on MipNeRF360 and clear visual improvements on Ref-Real.
- The proposed method is well motivated. While view-dependent specular effects are commonly represented as view-independent material properties, the authors observe that Gaussian Splatting inherently produces view-consistent appearance. Their proposed modules effectively address this, leading to improved performance.
- The paper is well-written and structured.

Weaknesses:
- Further verification and clearer explanation of Equation (1) should be provided, particularly regarding the undefined "screen-space probability" term.
- Additional ablation studies on window size parameters (particularly for pixel patch size in Multi-view Material Consistency loss and Reflection Strength Prior calculations) are welcome.
- It is unclear whether 3DGS or 2DGS serves as the baseline (inconsistent in the abstract and the Normal Prior sections with other parts).
- Figure 3 is a little bit small, and it is suggested to provide clearer annotations to enhance the interpretability.

---

> ### Author Rebuttal · Authors · 2025-07-30
>
> Dear reviewer, thanks for your valuable comments and recognizing our **_impressive results_**, **_novel motivation_**, and **_well-organized structures_**. We are glad to provide further clarifications and experiments to address your concerns.
>
> **1. Explanation of Eq. (1).**
>
> Eq. (1) describes how Gaussian ellipsoids in 3D space are rasterized into 2D images, following the classic volume rendering formulation[1][2],
>
> $$
> \hat{\Psi} = \sum_{i=1}^N \psi_i * o_i * p_i * \prod_{j=1}^{i-1}(1-o_j).
> $$
>
> Here, $\psi_i$ denotes a selected attribute of the Gaussian $G_i$, such as diffuse color, albedo, and metallic. $o_i$ represents the opacity of $G_i$. The term $p_i$ indicates the probability that a given pixel lies within the projection of the 3D-space Gaussian onto the 2D screen space. The value of $p_i$ reaches 1 at the center of the projected Gaussian and decays outward. It is equivalent to evaluating a 2D Gaussian distribution function at the given pixel location. The term $\prod_{j=1}^{i-1}(1-o_j)$ denotes the accumulated transmittance, indicating how much light passes through the preceding Gaussians before reaching $G_i$. We will add a more clear clarification in the revised version.
>
> **2. Ablation studies on patch size.**
>
> The patch size refers to the side length of the image patch warped from the reference view to the source view. We conduct additional ablation studies on the effect of patch size in Multi-view material consistency and Reflection strength prior on Ref-Real dataset, as reported below. As shown in the table, different patch sizes exhibit similar results, except when the patch size is too small, which makes the patch error more sensitive to outliers. These results demonstrate our robustness to the selection of patch size. In our implementation, we use path size as 7 for both multi-view material consistency and reflection strength prior. We also notice that there is a typo in line 182 of the main paper, where we mistakenly wrote the patch size as 3×3 instead of 7×7. We will correct it in the revised version.
>
>
> |  | **Multi-view** | **material** | **consistency** /// | **Reflection** | **strength** | **prior** |
> |-------------|----------------|----------------|-------------------|----------------|----------------|-------------------|
> | **Patch Size** | **PSNR$\uparrow$** | **SSIM$\uparrow$** | **LPIPS$\downarrow$** | **PSNR$\uparrow$** | **SSIM$\uparrow$** | **LPIPS$\downarrow$** |
> | 1           | 23.3989        | 0.6408         | 0.1900            | 23.4358        | 0.6434         | 0.1875            |
> | 3           | 23.4237        | 0.6422         | 0.1895            | 23.4372        | 0.6437         | 0.1871            |
> | 7           | 23.4378        | **0.6439**     | 0.1870            | 23.4378        | 0.6439         | **0.1870**        |
> | 11          | 23.4375        | 0.6438         | 0.1871            | **23.4380**    | **0.6441**     | **0.1870**        |
> | 15          | 23.4347        | 0.6437         | **0.1868**        | 23.4377        | 0.6439         | **0.1870**        |
> | 19          | **23.4389**    | 0.6435         | 0.1873            | 23.4375        | 0.6438         | 0.1871            |
>
> **3. 3DGS or 2DGS.**
>
> We exactly use 2DGS as our representation. We will carefully review the paper and ensure consistent description.
>
> **4. About Figure 3.**
>
> We provide a more clear clarification of the pipeline of Figure 3 in the following. We will enlarge Figure 3 and add symbols alongside their corresponding textual descriptions in the revised version to improve interpretability.
>
> &nbsp;&nbsp;i) Select a reference view and M source views.
>
> &nbsp;&nbsp;ii) Warp pixel patches from reference view to source views. (Eq. 4)
>
> &nbsp;&nbsp;iii) Compute reflection scores between patches on GT RGB images. (Eq. 6, Fig. 3a)
>
> &nbsp;&nbsp;iv) Back-project the reflection scores into a 3D point cloud and compute the reflection strength prior. (Fig. 3b)
>
> &nbsp;&nbsp;v) Use rendered metallic maps to compute $\Gamma$ and $M_0$. (Fig. 3c)
>
> &nbsp;&nbsp;vi) Use the reflection strength priors $w_{ref}$, $\Gamma$, $M_0$ to compute $L_{ref}$. (Eq. 7)
>
> [1]. Robert A Brebin, Loren Carpenter, and Pat Hanrahan. Volume rendering. In Seminal graphics: pioneering efforts that shaped the field, pages 363–372. 1998.
>
> [2]. Bernhard Kerbl, Georgios Kopanas, Thomas Leimkühler, and George Drettakis. 3D Gaussian Splatting for Real-Time Radiance Field Rendering. ACM Trans. Graph., 42(4):139–1, 2023.

---

> > ### Comment · Reviewer_t1W5 · 2025-08-05
> >
> > Thanks the authors for the responses. The clarifications make the paper clearer to me. I hope the authors can include the details and additional ablation studies in the final version.

---

> > > ### Author Response · Authors · 2025-08-05
> > >
> > > Dear Reviewer,
> > >
> > > We are glad to know that our rebuttal has addressed your concerns, and we truly appreciate your positive and thoughtful feedback. In the final version, we will make sure to include additional discussions and clarifications, including implementation details, equation explanations, and ablation studies.
> > >
> > > Best,
> > >
> > > The authors

---

### Official Review · Reviewer_LbeZ · 2025-07-02

**Clarity:** 3
**Significance:** 3
**Originality:** 3
**Rating:** 4
**Confidence:** 4

**Summary:**

This paper tries to address the problem of inverse rendering with highly specular objects. Specifcally, it will reconstruction a scene with geometry represented with 2D Gaussians, and BRDF of each 2D Gaussians and an enviromentmap lighting from multiview posed images.

This problem has been long studied. The major novel techniques introduced in this paper is the multiview material consistency and reflection strength prior. Other techniques like visibility mask, normal loss, indirect lighting and split-sum approximation are not novel but it's non-trival to make it work and integraated into the proposed system.

The author report novel view synthesis results on ShinyBlender, GlossySynthetic, Ref-Real and Mip-NeRF360, achieving state-of-the-art results especially on ShinyBlender, GlossySynthetic and Ref-Real.

**Questions:**

A video demo of either relighting or novel view synthesis would be great!

Specularity is better visualized through videos of changing viewpoints or videos of changing env lighting. In the supp, I only find videos of changing viewpoints for a few specular objects, it would be great if you can also have a video of changing lighting.



The methods are a little bit complicated, it will be great to have a pseudocode for the major component of it.  Also, a quick question, the code for computing multiview material consistency and reflection prior could largely be the same?  if so, that would ease the cost of engineering.

About evaluating extracted environment map lighting or extracted specular and diffuse colors. I don't think this is necessary, but it's very good to have, especially about the lighting.

**Ethical Concerns:**

["NO or VERY MINOR ethics concerns only"]

**Final Justification:**

I read the author's rebuttal and it address most of my concerns. Given the promise that the authors will provide video visualization of changing lighting, I retain my score as weak accept.

**Limitations:**

Yes

**Quality:**

3

**Strengths And Weaknesses:**

Strength:

The results looks very impressive. The novel view rendering results, and the extrated geometry and also the visualization of the extracted environment map looks just expressive.

The author also ablated the complicated system design, especially the multiview material consistency and reflection prior.



Weakness:

Overall, I don't think there are some major weak point. My only feeling is that the proposed methods is a little bit too complicated.

---

> ### Author Rebuttal · Authors · 2025-07-30
>
> Dear reviewer, thanks for your valuable comments and recognizing the **_impressive results_** and the **_novelty_** of our work. We are pleased to provide further clarifications and additional experiments to address your concerns.
>
> **1. About video visualization.**
>
> In the supplementary video, we provide novel view synthesis results on both synthetic and real scenes, material decomposition results, and material editing results in terms of diffuse color and roughness. We also provide novel view synthesis of changing environment maps for relighting at the end of the video. Unfortunately, due to the restrictions on rebuttals, we are unable to show more environment relighting results. As for additional relighting applications such as inserting extra light sources (point lighting or parallel lighting), we can render these results based on our inferred material decomposition. We will include more visualization results in the revised version of our video.
>
> **2. About complexity of our method.**
>
> We provide a flowchart for the two regularization strategies introduced in Section 3.2 to make it more clear.
>
> &nbsp;&nbsp;&nbsp;&nbsp;**(I) Multi-view Material Consistency.**
>
> &nbsp;&nbsp;&nbsp;&nbsp;&nbsp;&nbsp;i) Select a reference view and M source views.
>
> &nbsp;&nbsp;&nbsp;&nbsp;&nbsp;&nbsp;ii) Warp pixel patches from reference view to source views. (Eq. 4)
>
> &nbsp;&nbsp;&nbsp;&nbsp;&nbsp;&nbsp;iii) Compute $L_{mv}$ between patches on rendered material maps. (Eq. 5)
>
> &nbsp;&nbsp;&nbsp;&nbsp;**(II) Reflection Strength Prior.**
>
> &nbsp;&nbsp;&nbsp;&nbsp;&nbsp;&nbsp;i) Select a reference view and M source views. (The same as above)
>
> &nbsp;&nbsp;&nbsp;&nbsp;&nbsp;&nbsp;ii) Warp pixel patches from reference view to source views. (Eq. 4) (The same as above)
>
> &nbsp;&nbsp;&nbsp;&nbsp;&nbsp;&nbsp;iii) Compute reflection scores between patches on GT RGB images. (Eq. 6, Fig. 3a)
>
> &nbsp;&nbsp;&nbsp;&nbsp;&nbsp;&nbsp;iv) Back-project the reflection scores into a 3D point cloud and compute the reflection strength prior. (Fig. 3b)
>
> &nbsp;&nbsp;&nbsp;&nbsp;&nbsp;&nbsp;v) Use rendered metallic maps to compute $\Gamma$ and $M_0$. (Fig. 3c)
>
> &nbsp;&nbsp;&nbsp;&nbsp;&nbsp;&nbsp;vi) Use the reflection strength priors $w_{ref}$, $\Gamma$, $M_0$ to compute $L_{ref}$. (Eq. 7)
>
> As the reviewer observed, the i) - iii) steps of both processes share a very similar logic, which allows us to use the same implementation in practice. The only difference is which maps are applied to compute the patch errors.
>
> **3. Evaluation of environments and materials.**
>
> It is not easy to directly evaluate the predicted diffuse and specular colors in these benchmarks, as there are no ground truths available for reference. As an alternative, we evaluate the performance of the material decomposition on Synthetic4Relight dataset which provides ground truth materials, as reported in Tab. 1 in supplementary materials. We further evaluate the accuracy of the learned environment maps on ShinyBlender dataset, as reported below. The results demonstrate that our multi-view consistent material decomposition and environment modeling contribute to learning more accurate environmental illumination.
>
> |        | GaussianShader | 3DGS-DR | Ref-Gaussian | Ours |
> |--------|----------------|---------|--------------|------|
> | **LPIPS↓** | 0.656          | 0.543   | 0.495        | **0.406** |

---

> > ### Comment · Reviewer_LbeZ · 2025-08-03
> > **Thanks for the rebuttal, addressed most of my problems**
> >
> > I read the rebuttal, it answered most of my questions. Due to rebuttal limitations, the author could not provide video visualization of changing lighting but promised to include in future version.
> >
> > I retain my score as Borderline Accept.

---

> > > ### Author Response · Authors · 2025-08-04
> > > **Thanks for the accept recommendation**
> > >
> > > Dear reviewer LbeZ,
> > >
> > > Thanks for your time and expertise. We are glad to know that our rebuttal addressed your concerns. We really appreciate the accept recommendation. We will include more video visualization of changing lighting in the final version.
> > >
> > > Best,
> > >
> > > The authors

---

### Author Response · Authors · 2025-08-04
**We will be happy to take questions**

Dear reviewers,

Thank you very much for taking the time to provide your valuable comments and suggestions. We sincerely appreciate your feedback and hope that our rebuttal addresses your concerns. Please do not hesitate to let us know if there is anything we can further clarify. We would be happy to take this opportunity to discuss with you.

Thanks,

The authors

---

### Note · Authors · 2025-08-12

We thank the reviewers for recognizing the impressive results, novel motivation, and well-structured presentation of our work. During the rebuttal, we provided additional experiments and clarifications to address the reviewers’ concerns regarding implementation details, ablation studies, visualizations, and other aspects. We are happy to know that all reviewers indicated their concerns have been resolved. We will incorporate the visualizations, experiments, and implementation details discussed in the rebuttal into the final version. We sincerely appreciate the reviewers for taking the time to evaluate our paper and for providing valuable feedbacks.

---

### Decision · Program_Chairs · 2025-09-17

**Decision:**

Accept (poster)

**Comment:**

This paper presents MaterialRefGS, a novel framework that significantly improves the rendering of reflective scenes in Gaussian Splatting by introducing multi-view material consistency constraints and a reflection strength prior. Reviewers were impressed by the state-of-the-art visual quality and the novelty of the core ideas. The authors provided an exceptionally detailed rebuttal with new quantitative experiments and visualizations that successfully addressed all initial reviewer concerns, leading to a clear and unanimous consensus for acceptance.